# Survey evaluation of dog owners' feeding practices and dog bowls' hygiene assessment in domestic settings

Emily Luisana[1☯¤a]*, Korinn Saker[1☯], Lee-Ann Jaykus[2‡], Caitlyn Getty[1¤b‡]

**1** Department of Molecular Biomedical Sciences, College of Veterinary Medicine, North Carolina State University, Raleigh, North Carolina, United States of America, **2** Department of Food, Bioprocessing and Nutrition Sciences, North Carolina State University, Raleigh, North Carolina, United States of America

☯ These authors contributed equally to this work.
¤a Current address: BluePearl Veterinary Specialty Hospital, Cary, North Carolina, United States of America
¤b Current address: NomNomNow Inc, Nashville, Tennessee, United States of America
‡ LAJ and CG also contributed equally to this work.
* ektownse@gmail.com

**Data Availability Statement:** Most data are in the Supporting information files. Some data cannot be shared publicly because of privacy concerns (such as names and email addresses). Data are available from the Institutional Research and Planning

## Abstract

In-home pet food handling and food dish hygiene practices can have adverse health impacts for both humans and pets. Safe food and dish handling guidelines are not easily evidenced for pet owners. The study was designed to investigate dog owners' feeding habits and evaluate the impact of the Food and Drug Association (FDA) hygiene protocols on dog food dish contamination. Procedures and surveys were approved by North Carolina State University Institutional Animal Care and Use Committee and Institutional Review Board. Pet feeding and food dish hygiene data were collected from 417 dog owner surveys and 68 food dish swabs. Total aerobic plate counts (APC) were performed on 68 dishes and randomly assigned into Group A (FDA pet food handling and dish hygiene guidelines), Group B (FDA pet and human food handling and dish hygiene guidelines), or Group C (no guidelines). Hygiene protocols were instituted in-home for 1 week, followed by a second APC and follow-up survey. Survey from dog owners-households indicated: 4.7% were aware of FDA pet food handling and dish hygiene guidelines; 36% have individuals $\leq$ 13 years old and/or immunocompromised; 43% stored dog food 0–5 feet from human food; 34% washed their hands after feeding; and 33% prepared their dog food on human food preparation surfaces. The hygiene protocols followed by Groups A and B resulted in significant decreases in food dish APC (p<0.001; 1.4; (0.9, 2.0); p<0.05; 0.604 (0.02, 1.2), respectively), as compared to Group C (p$\geq$0.05). Hot water (>160˚ F or 71.1˚C) washing decreased APC (p<0.01; 1.5 (0.4, 2.6)) over cold/lukewarm water. In the follow-up survey, 8% of Group A and B respondents reported likely to adhere to protocols long-term. This study suggests a need for pet food handling and dish hygiene guideline education to minimize bacterial contamination of dishes, especially for high-risk populations.

committee (contact via Nancy Whelchel, PhD) for researchers who meet the criteria for access to confidential data.

**Funding:** The authors received no specific funding for this work.

**Competing interests:** The authors have declared that no competing interests exist.

## Introduction

The Centers for Disease Control and Prevention (CDC) One Health initiative highlights the interconnection between humans, animals, and the environment. The aim of One Health is ultimately to achieve optimal health outcomes for all involved in these interactions [1]. Food safety-related concerns are one aspect of One Health that span contamination of human and animal foodstuffs as well as equipment and environment hygiene practices involved in food handling. Certainly, human food safety is paramount to human wellness; correspondingly, emphasis on benefits of enhancing the human-animal bond invites One Health concerns for pet food safety. Pet feeding involves interplay between the pet, the owner, and the food. This interaction creates the opportunity for mutual exchange of microbial contaminants from food or water, dishes, and the food storage or preparation environment, which can cause health consequences for both humans and pets.

Drug-resistant *Escherichia coli* has been demonstrated to be present on pets, humans, and the pets' food dish in affected households [2]. A 2006 study examining microbial contamination, measured by total aerobic plate counts, of daily use objects in households found that pet food dishes had the ninth highest level of contamination, out of 32 household surfaces studied [3]. The same study performed bacterial cultures of the household objects for medically important species: methicillin-sensitive *Staphylococcus* spp. (MSSA, found in 15% of pet food dishes sampled), methicillin-resistant *Staphylococcus aureus* (MRSA, 3%), coagulase-negative *Staphylococcus* spp (74%), pseudomonads (18%), and Enterobacteriaceae (36%). In 2010, Weese and co-workers isolated *Clostridium difficile* in 6 of 84 dog food dishes making it one of the second most contaminated sites of those sampled, ranking higher than surfaces commonly considered to have high bacterial loads such as the toilet [4]. More recently, a 2012 study also examining total aerobic counts on household surfaces showed that pet water dishes had the third-highest bacterial counts out of 26 surfaces studied. When categorized into areas of the household, the category of pet-related items (which comprised the water dish and a pet toy) had the highest bacterial counts [5]. These studies corroborate the concern of dog dishes being a potential source of microbial contamination in a household setting.

Despite the concern for contamination, few guidelines for pet dish hygiene exist and those guidelines are not easily accessed or widely distributed. The U.S. Food and Drug Administration (FDA) has pet dish cleaning recommendations available via their website in combination with general pet food handling guidelines [6], but in comparison to their guidelines for human dishes in the FDA Food Code 2017 [7], the pet information is sparse and vague. In addition, no studies examining the effects of the FDA's recommendations on pet dish hygiene were found by the authors. Therefore, the goals of the study were to assess: dog owner's awareness of FDA pet food handling and feeding dish hygiene guidelines; pet food and dish handling habits of pet owners; and evaluate the degree of dog bowl bacterial contamination before and after the implementation of the FDA pet food guidelines and FDA Food Code guidelines. The authors hypothesized that awareness and compliance of FDA guidelines would be low and that both the FDA pet food and Food Code guidelines would result in a significant decrease in dog bowl contamination.

## Materials and methods

### Survey study design

Study procedures were approved by the North Carolina State University's Institutional Animal Care and Use Committee (IACUC protocol number 19–542) and Institutional Review Board (IRB protocol number 23476).

Study participants were dog owners recruited from local veterinary practices, social media, and university veterinary school staff and faculty. Recruitment criteria included owning at least one dog who eats from a designated food bowl. An incentive in the form of a dog food donation to a local shelter for each participant was offered. The approximately 20-min survey was powered by Qualtrics and developed with the assistance of an experienced psychometrician. The survey was developed to obtain information regarding their pet's signalment, health status, and diet. Additionally, information was obtained regarding the extent of each pet owner's knowledge of current FDA-published pet food handling and food bowl hygiene guidelines and the specific food handling and bowl hygiene habits practiced at home for their dog(s). Owners were requested to complete one survey per dog with a maximum of two surveys per family. A total of 417 surveys were returned. Qualtrics surveys were evaluated via internal software for means and tabulated breakdown.

## Evaluating food bowl bacterial contamination

The impact of following specific food handling and food bowl hygiene protocols on food bowl contamination risk was evaluated using a subset of the survey participants. From the survey participants, owners of 68 dogs (a total of 50 owners) were invited to complete a food bowl bacterial contamination study. To minimize bias, prior to survey distribution to these 50 dog owners, a baseline food bowl swab was obtained, then owners were asked to complete the Qualtrics survey. Participants were then randomly assigned to three treatment groups. Treatment group A (n = 27) were instructed to follow the FDA's Tips for Safe Handling of Pet Food and Treats [6] (last update 7/9/2019 at the time of study). Specifically, they were requested to: wash their hands before and after handling pet food, to not use their dog food bowl as a food scooping utensil, wash the bowl and scooping utensils with soap and hot water after each use, discard uneaten food in a designated manner and store dry pet food in its original bag. Treatment group B (n = 30) were given the FDA's Tips for Safe Handling of Pet Food and Treats and more stringent instructions extrapolated from the FDA's Food Code 2017 [7]. These instructions specified that handwashing should be at least 20 seconds and with soap and warm water, food dishes should be scraped of food prior to washing, dishes should be washed with soap and water >160˚ F (71.1˚C) for at least 30 seconds and dried thoroughly with a clean towel or put through a National Sanitation Federation (NSF) certified dishwasher for a wash and dry cycle. Treatment group C (n = 11) was given no specific instructions regarding food and or bowl handling but informed of the second sample collection time. Owners were asked to follow the specific protocol until the second bacterial swab of the pet's food bowl was obtained (average of 8 d following pre-protocol sample). A follow-up survey was sent to Groups A and B regarding their compliance and impression of the given instructions. The group C follow-up survey focused on their food bowl washing behavior since the baseline sample was taken.

To obtain the food bowl swabs for evaluation of bacterial contaminants, bowls were fitted with a measured 10 cm$^2$ environmental sampling template (SKC Incorporated, Eighty Four, PA USA) to allow for accurate plate counts. If bowls were too small for the standard template the contact surface area was measured. A sterile swab, saturated in 4 mL of Butterfields solution (Puritan ESK pre-filled Environmental Sampling Kit, Guilford, ME USA), was systematically rolled across the bowl surface surrounded by the template, then placed back into the solution and sealed for proper handling. Samples were kept on ice and plated on aerobic Petrifilm (3M Nelson Jameson Company, Marshfield, WI USA) within 24 h of collection. Samples were plated at 0, 1:10, 1:100, and 1:1000 dilutions. Swab sampling and plating were performed by one investigator and repeated at the end of the treatment period. Post samples for Groups A and B were

plated at 0 and 1:10 dilutions only based on suspected contamination level post-treatment and the results of a pilot study. Following manufacturer's instructions, the Petrifilm was incubated for 48 h +/- 1 h at 32˚C. Total aerobic plate counts (APC) were then read manually and results were adjusted to account for sample surface area on the basis of colony-forming units per cm squared (CFU/cm$^2$). Results were evaluated using R version 3.6.2 [8]. For log-scale analyses, all raw values were increased by 0.01 to allow the logarithm transformation to be applied to the 0 values. Additionally, the logarithm with base 10 ($log_{10}$) was used for the transformation. For percent reduction evaluations, observations with a 0-value for the pre-value were excluded (2 of 68 observations). Predictive models of the change in log-transformed plate counts were examined with linear regression and Kruskal-Wallis tests. Linear models were fit with independent variables consisting of survey questions and group assignment along with their interactions. The dependent variable was always a transformation change in CFUs or total CFUs pre or post. Model selection was done in a backward manner with ANOVA p-values as the determiner of removal with a p-value greater than 0.10 indicating a consideration of removal. The Kruskal-Wallis test was used to examine non-normal outcomes, including percent reduction and total plate growth, as they varied based on treatment group or groups based on initial growth values. T-tests were performed within groups to compare pre- and post-treatment values. Confidence intervals (95%) are provided in parentheses after estimates for significant differences.

## Results

### Survey

A total of 417 surveys were returned. As not all questions were required, there was a range in total responses to individual questions. There was a broad dog demographic represented in this study as reported by survey responses from dog owners. Reported age ranged from < 12 months to 16 years with an average of 7 years. Gender distribution was as follows: spayed female (43%), neutered male (41%), male intact (11%), female intact (5%). The majority (44%) reported breed as mixed or 'other' breed. The most popular purebred dog reported was a Labrador Retriever at 9%, followed by German Shepherd Dog at 4%. Body weight distribution was as follows: 1–10 pounds (0.5–4.5 kg, 5%), 11–25 pounds (5–11.4 kg, 20%), 26–50 pounds (11.8–22.7 kg, 25%), 51–75 pounds (23.2–34 kg, 29%), 76–100 pounds (34.5–45.5 kg, 16%), 101+ pounds (45.0+ kg, 5%). The majority of respondents (76%) reported their dogs as healthy, whereas 24% reported a history of illness: gastrointestinal-related food allergies (7%), dental disease (6%), obesity (5%), pancreatitis (2%), liver disease, bladder stones, kidney disease and unspecified neoplasia (less than 2% each).

A minority, less than 5%, of respondents were aware of the existence of FDA pet food handling guidelines. However, when asked where they expected to find this information, 8% replied the FDA, 41% the food label, 28% their veterinarian, 11% the store of purchase, 6% the USDA and 6% various websites. Table 1 summarizes the dog owner compliance in our study. Over 75% of respondents reported compliance regarding: inspecting packaging for visible damage, avoiding use of the food bowl as a scooping utensil, tightly covering leftover pet food, discarding food in a way a pet cannot access, and avoiding raw food. Under 25% of respondents reported compliance regarding: washing hands as recommended prior to handling pet food, washing the food dish as recommended after each use, and washing the food scoop as recommended after each use.

The majority of respondents (22%) reported washing their dish, on average once weekly. However, there was a wide distribution of responses with 12% reported washing their dish at least once daily to 18% reported they wash their dish either less than every 3 months or not at all.

**Table 1. FDA pet food handling recommendation and owner reported compliance.**

| FDA Pet Food Handling Recommendation | Owner Reported Compliance |
|---|---|
| Inspect for visible damage | 86% |
| Wash hands with soap and hot water for at least 20 seconds *prior* to handling | 22% |
| Do not use bowl as scooping utensil | 91% |
| Wash pet food *dish* with soap and hot water after each use | 50% washed with hot/water or dishwasher |
| | 12% washed at least once daily |
| Wash *scoop/utensil* with soap and hot water for at least 20 seconds after each use | 13% |
| Wash hands with soap and hot water *after* handling | 38% |
| Store food in original bag | 30% (including those who put whole bag into larger container) |
| Tightly cover leftover food | 81% (dry food) |
| | 57% (canned food) |
| Discarding food in a way pet cannot access | 96% |
| Do not feed raw food | 97% |

When respondents did wash their bowl, it was most often with soap and warm water (temperature defined as 100–159°F or 37.8–70.6°C, 36%) followed by the dishwasher (33%), soap with hot water (temperature defined as ≥160°F or >71.1°C, 17%), rinsing with water only (6%), soap with cool water (5%) with the remainder (<3%) reporting undefined average protocols. Most reported allowing their dish to air dry (44%), followed by hand-drying with a towel (32%), heated dry in a dishwasher (22%) and a smaller percentage used a non-heated dry in a dishwasher (<3%). When washing their pet food bowl, some respondents reported washing alongside human dishes (Table 2). The majority of respondents (65%) remove dry dog food from the manufacturer's bag for storage. Most respondents (81%) typically tightly closed or

**Table 2. Additional survey questions.**

| Additional Survey Questions | Owner Response |
|---|---|
| Where do you typically prepare your dog's food? | On a surface used for human food preparation (32%) |
| | Not on a surface used for human food preparation, but in the same room (39%) |
| | In a different room from where human food is prepared (29%) |
| When you wash your pet's food dish, do you wash it in the same sink/dishwasher used for human dishes? | Yes, it is washed with human dishes (43%) |
| | Yes, although it is washed separately from human dishes (49%) |
| | No, it is washed in a different sink/dishwasher than used for human dishes (8%) |
| Where do you typically keep your dog food dish? | Indoors (96%) |
| | Outdoors (4%) |
| If you had questions regarding how to handle or store your pet's food, where would you expect to find guidelines? (choose all that apply) | The pet food label (41%) |
| | Your veterinarian (28%) |
| | Place of purchase (11%) |
| | FDA (8%) |
| | USDA (6%) |
| | Other (6%, most common fill in answer: internet searches) |

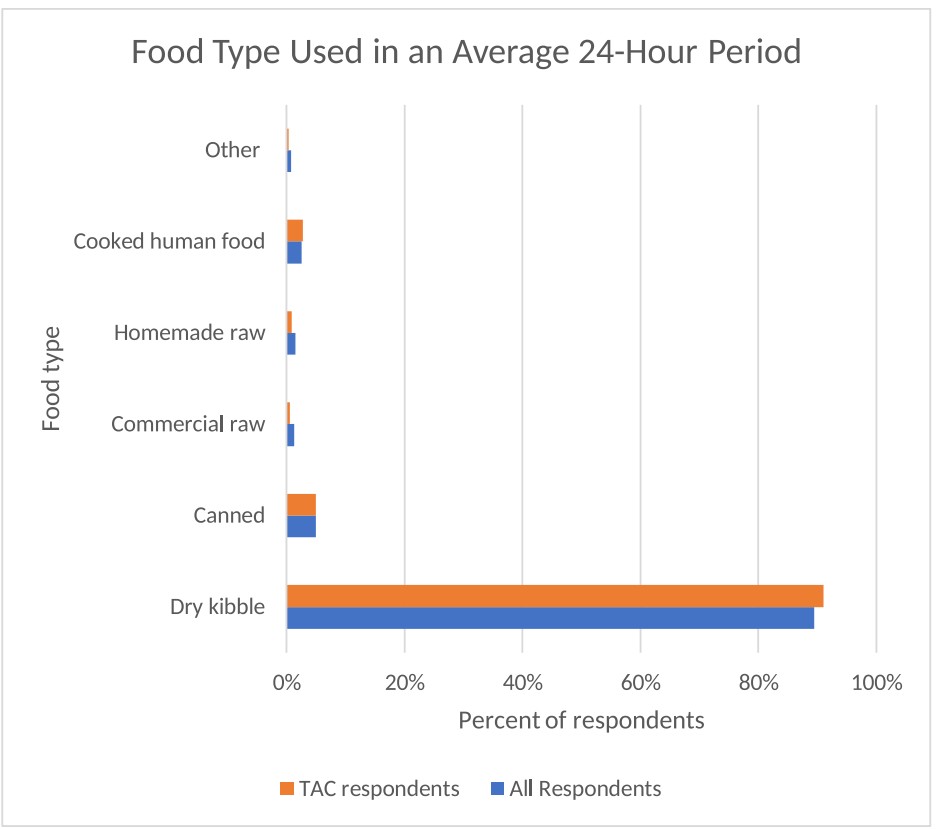

**Fig 1. Food type used in average 24-h period.** Comparison of food type among all survey respondents and respondents of those participating in the APC assessment.

sealed the bag/container in which the kibble is stored. Of those that fed canned food, most (61%) stored leftover food in the can, and (57%) reported using an airtight cover.

Other results of potential public health interest were not directly related to FDA recommendations. Roughly two-thirds of respondents reported either preparing their pet's food on a surface used for human food preparation or in the same room (Table 2). Regarding pet food storage, 44% reported storing dog food 0–5 feet from human food. Questions pertaining to the human population of the household found that 35% stated they have children <12 years old and/or immunocompromised individuals in the household.

Respondents were asked about the percentages of the food type placed in their dog bowl during an average 24-h period (Fig 1). The APC group was a relatively accurate reflection of the overall group as the majority (91% APC, 90% overall) fed kibble with 5% of each fed canned food. A smaller number of respondents in both groups used other categories such as cooked homemade food (3% of each group), raw commercial food (1% of each group), and raw non-commercial food 1.54 (<1% APC, 3% overall). Within the 3% of the overall group who reported raw non-commercial food, 25% noted they fed raw meat or eggs, 49% raw vegetables, 15% raw fruit, and 3% raw dairy. The bowl material for each group was also comparable (Fig 2) with the majority of each group being metal (74% APC, 64% overall), followed by plastic (16% APC, 19% overall), ceramic (10% APC, 16% overall) and ~1% of both groups reporting glass or other materials. Of the overall group, 9% reported adding supplements or medications into their pets' food bowls within the past 24 h.

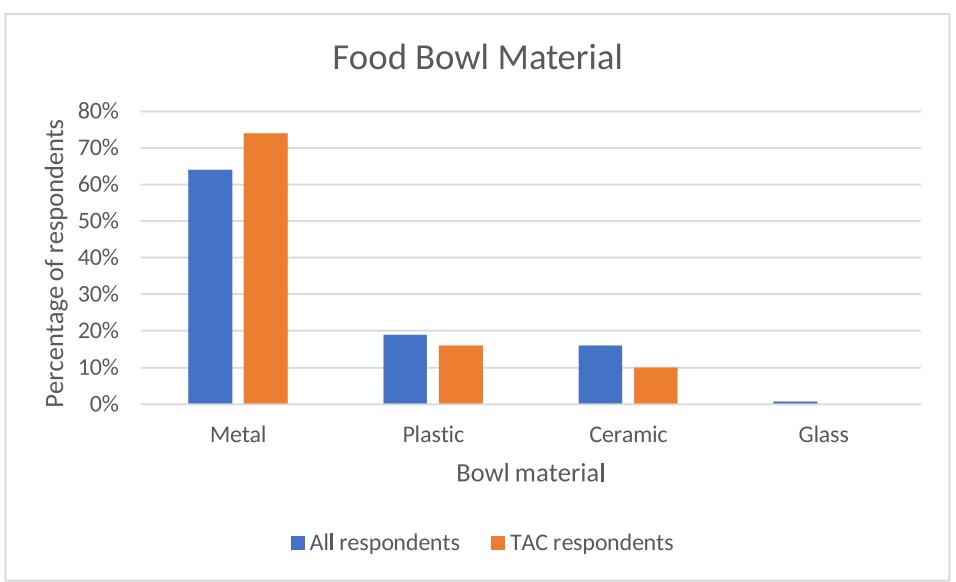

**Fig 2. Food bowl material.** Comparison of bowl type among all survey respondents and respondents of those participating in the APC assessment.

### Bacterial contamination evaluation

As is common with APC, there was a wide range in values, therefore data were examined on a log10 scale. Differences were found within groups A and B for APC between the pre- and post-treatments. Group C showed no significant change in APC from the initial to the final bowl swab for APC (Fig 3).

A significant decrease in APC was observed between pre- and post-bowl hygiene treatments in both Groups A (1.5 (0.9, 2.0), p<0.001) and B (0.6 (0.02, 1.2), p<0.05). Whereas group C showed a non-significant (p≥0.05) increase in APC (Fig 3). Once data were corrected for multiple testing via Bonferroni's method, no significant difference (p≥0.05) was noted when comparing the absolute quantitative decrease in APC between pre- and post-treatment for group A vs group B. To address the observation that a wide range of APC values was counted in bowls pre-hygiene treatment, we evaluated the APC changes from the perspective of split levels (low = <20 CFUs, medium = 20–100 CFUs and high = >100 CFUs) based on the pre-treatment contamination values. This further delineation did not show a significant difference in post-measurements across groups A and B (p≥0.05). In addition, utilizing a linear regression model with log-transformed post-contamination levels as the response and pre-contamination levels as the predictor, no difference in post-treatment APC values was found based on pre-contamination levels (p≥0.05).

Bowl material did not have a significant effect on CFU values of the aerobic bacteria detectable by our APC technique prior to initiating any food bowl hygiene treatment (p≥0.05). As well, no significant change in APC was noted in treatment groups A and B (p≥0.05) following the specified hygiene treatment. Additionally, the pre-treatment APC did not differ based on the presence of immunocompromised individuals or children in the household (p≥0.05).

The follow up survey was completed by 90% of APC participants. Only 8% of Group A and B respondents reported likely to adhere to all of the instructed protocols long-term. This included handwashing, dishwashing, and food storage guidelines. On the other hand, 20% reported likely to follow only their given washing instructions long-term. Group C participants

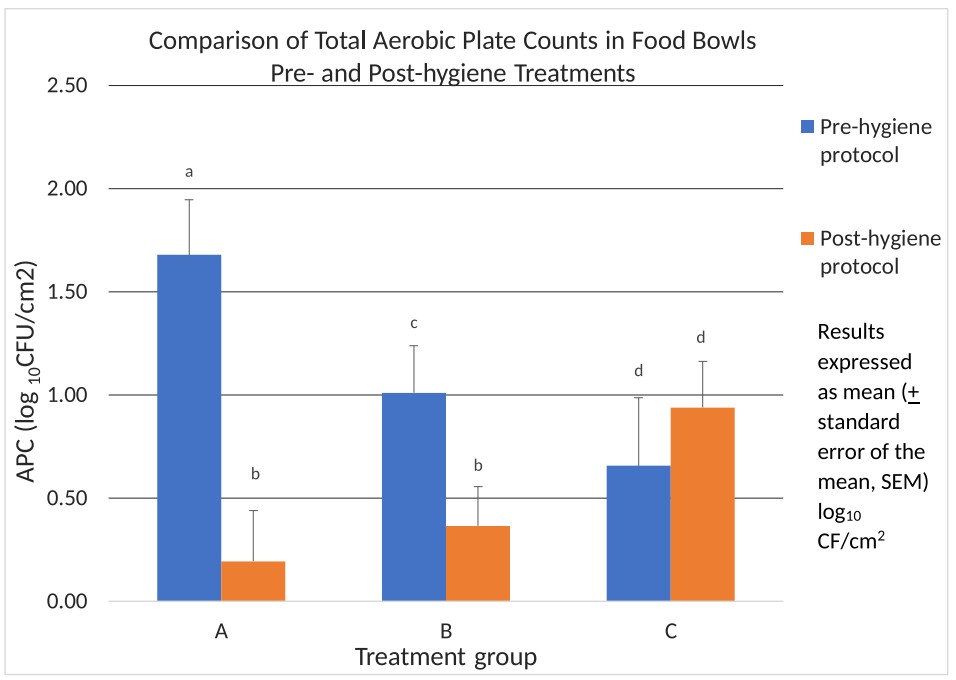

**Fig 3. Comparison of the total aerobic plate counts in studied dog bowls pre- and post- hygiene treatment.** APC on a basis of colony forming units (CFUs) per cm$^2$ of studied bowls pre- and post-hygiene treatment. Superscripts that differ within a group indicated significant differences (p<0.05). Similar superscripts across groups indicate no significant difference (p≥0.05).

were not given instructions, however; none had washed their bowl since the first sample was taken. No significant differences were found between groups A and B in the self-reported likelihood of continuing all instructions (p≥0.05) or washing instructions (p≥0.05).

A linear model predicting log-change in CFUs by washing method and drying method indicated that a significant difference, reflective of APC, was observed between cold/lukewarm wash and FDA recommended methods (dishwasher or hot water wash). The difference was a decrease of 1.5 units on the log scale (p<0.01; 1.5 (0.4, 2.6)) for APC following a hot water wash or dishwasher as compared to a cold/lukewarm water wash. No significant effect of the drying method was found within or across any treatment group (p≥0.05).

## Discussion

It was found that the vast majority of study dog owners were not aware of and did not follow FDA pet food handling and storage guidelines. Response to individual recommendations varied, however hygiene-related handling practices (washing of hands, bowl and utensil) showed overall low levels of compliance. Additionally, studies in humans regarding self-reported handwashing show an overestimation of hygiene [9] and similar forces, including the effects of social desirability bias, could be expected in this study. Exposure to contaminated dog food can have implications for canine and human health. For example, there have been multiple outbreaks of both humans and dogs becoming ill after exposure to dog food contaminated with pathogenic bacteria [10]. These risks may be amplified in households with children and/ or immunocompromised individuals, which were over a third of respondents' households. The preponderance of pet food recalls has heightened the awareness of risk of illness. The CDC's examination of a 2008 multi-state dog food recall found that the attack rate supported

hypothesized transmission methods regarding pet food handling, including cross-contamination in the kitchen and irregular cleaning of dog food dishes [11]. Microbial contamination has been reported in kibble [11] and the increasing prevalence of both commercial and homemade raw diets, which carry an increased risk of microbiological contamination such as *E. coli* and *Salmonella* [12], exacerbates these concerns. According to Weese et al. (2010), dog bowls were 17 times more likely to be contaminated with *Clostridium difficile* if the dog was fed a commercial raw food diet compared to other types of food [4]. These diets can involve increased preparation within the kitchen environment, which may further increase human exposure. Risks also exist outside of food contamination; the aforementioned study investigating household spread of drug-resistant *E.coli* hypothesized that the bacteria found in one dog bowl originated from the pet's feces [2].

However, the risk of contamination of the household can be mitigated. We concluded that bacterial contamination is impacted by dish washing protocols due to the significant decrease in APC for both Groups A and B, but not in Group C. Although this study did not differentiate between pathogenic and non-pathogenic bacterial species, APC are commonly used in the food industry to determine the efficacy of sanitation. The CDC's cleaning and sanitization guidelines for human dishes are based on achieving a 5-log reduction in bacterial counts [13]. The degree of contamination of bowls in this study did not allow for an assessment of sanitation by this definition; however, the significant reduction in APC in both Group A and B showed a beneficial impact of following either protocol. As these protocols each had multiple steps, further studies identifying the best methods for sanitizing dishes are needed. However, as only 20% of Group A and B respondents reported they were likely to follow their hygiene instructions long-term, and only 8% reported likely to follow all given instructions, the need for recommendations that are feasible as well as effective should be emphasized. Studies should address potential concerns such as the effects of biofilms, the influence of bowl degradation on contamination, and the risk of cross-contamination in dishwashers. This is particularly true for pathogenic bacteria of high zoonotic potential. A 2006 study by Weese et al in which food dishes were inoculated with *Salmonella*-containing raw meat showed persistent contamination in the majority of pet dishes after washing with routine measures including a dishwasher or with soap and water [14]. Other studies have found dishwashers can disperse and harbor bacteria [15, 16]. Concerns regarding cross-contamination may extend beyond bacteria when one considers that 9% of pet owners reported adding medications or supplements into their pets' food bowls.

The majority of respondents reported storing their pet food against FDA and most manufacturers' recommendations, which may have implications as far as increased risk of microbial contamination [17], nutritional degradation [18] and palatability. In addition, some respondents were engaging in behaviors that may increase risk of bacterial contamination that were not addressed in FDA guidelines such as the location of food preparation and storage. It is noted that the FDA has added more specific recommendations to their website regarding pet food storage and pet food recalls (website updated 4/14/2020); however, it is not comprehensive in addressing pet owners' food preparation choices. Additionally, because survey respondents indicated low levels of awareness that the FDA was a source of such dog feeding hygiene recommendations, the expected sources of this information including the pet food label, veterinarians and pet food retailers, should consider prominently featuring these public health recommendations for their clients and/or customers. Further, it was noted Group C showed no significant change in APC, despite the survey and the knowledge of the upcoming sample collection serving as potential introducers of bias. This suggests that education beyond awareness is needed to allow for effective hygiene changes.

Sample size was a limitation to this study, in particular for subgroups such as raw diets. Future studies should further examine contamination with specific pathogenic bacterial species and consider the contamination risk of other microbiological agents or toxins. Finally, further studies identifying ideal cleaning and storage recommendations as well as best practices to communicate these recommendations to consumers would help minimize risk of microbial contamination in pet food after distribution as well as minimize health consequences to both pets and their human households.

## Supporting information

**S1 Table. APC data.**
(XLSX)

**S1 Dataset. Survey data-initial.**
(XLSX)

**S2 Dataset. Survey data- follow up.**
(XLSX)

## Author Contributions

**Conceptualization:** Emily Luisana, Korinn Saker, Lee-Ann Jaykus, Caitlyn Getty.

**Data curation:** Emily Luisana, Korinn Saker, Caitlyn Getty.

**Formal analysis:** Emily Luisana, Korinn Saker.

**Funding acquisition:** Korinn Saker.

**Investigation:** Emily Luisana, Caitlyn Getty.

**Methodology:** Emily Luisana, Korinn Saker, Lee-Ann Jaykus, Caitlyn Getty.

**Project administration:** Emily Luisana.

**Resources:** Emily Luisana, Lee-Ann Jaykus, Caitlyn Getty.

**Supervision:** Emily Luisana, Korinn Saker, Lee-Ann Jaykus, Caitlyn Getty.

**Validation:** Emily Luisana, Korinn Saker.

**Visualization:** Emily Luisana, Korinn Saker, Caitlyn Getty.

**Writing – original draft:** Emily Luisana.

**Writing – review & editing:** Emily Luisana, Korinn Saker, Caitlyn Getty.

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
