## [Decision Letter · Decision Letter 0]

24 Nov 2021

PONE-D-21-32996Survey evaluation of dog owners’ feeding habits in a household setting and comparison of FDA hygiene protocols on dog bowl bacterial contamination as evaluated by total aerobic cell counts.PLOS ONE

Dear Dr. Luisana,

Thank you for submitting your manuscript to PLOS ONE. After careful consideration, we feel that it has merit but does not fully meet PLOS ONE’s publication criteria as it currently stands. Therefore, we invite you to submit a revised version of the manuscript that addresses the points raised during the review process.

ACADEMIC EDITOR:The academic editor and the reviewers think that the present study is interesting and has merit to be published in PLOS ONE. Nonetheless, the authors are kindly requested to address the few comments raised by the reviewers and submit the revised manuscript for re-evaluation.

We look forward to receiving your revised manuscript.

Kind regards,

Alexandra Lianou, M.Sc., Ph.D.

Academic Editor

PLOS ONE

Journal Requirements:

Reviewers' comments:

Reviewer's Responses to Questions

**Comments to the Author**

1. Is the manuscript technically sound, and do the data support the conclusions?

Reviewer #1: Partly

Reviewer #2: Yes

2. Has the statistical analysis been performed appropriately and rigorously? 

Reviewer #1: No

Reviewer #2: Yes

3. Have the authors made all data underlying the findings in their manuscript fully available?

Reviewer #1: Yes

Reviewer #2: No

4. Is the manuscript presented in an intelligible fashion and written in standard English?

Reviewer #1: Yes

Reviewer #2: Yes

5. Review Comments to the Author

Reviewer #1: I can confirm that the subject matter of this study (Survey evaluation of dog owners’ feeding habits in a household setting and comparison of FDA hygiene protocols on dog bowl bacterial contamination as evaluated by total aerobic cell counts) is of interest and relevance for publication in PLOS ONE

Comments to the Authors:

- The authors did not present any hypothesis at the beginning

- add reference to statistical analysis

- correct temperature on the Celsius scale, the body weight - kg , add to weight ‘body’

In my opinion conclusion (summary) may be improved giving few key message/take home message to the readers. An idea may be to synthetize in 3-5 bullet the key results of the study, evidences and recommendation. This improvement will increase clearness and readability. Add a practical implications statement

Reviewer #2: The present study provides interesting information about dog owner’s pet food and dish handling habits as well as, dog owner’s awareness of relevant FDA guidelines. Additionally, this work assesses the impact of specific FDA food handling and food bowl hygiene protocols on food bowl contamination, as determined by aerobic plate count. The survey findings indicated that the majority of participants were not aware of FDA guidelines and hence they do not follow the indicated food handling practices. On the other hand, the dog owners who were requested to follow specific guidelines during the study, presented low levels of compliance to the instructed protocols long-term. However, the implementation of these protocols resulted in significant reduction in aerobic plate count between pre- and post-bowl hygiene treatments, while the dish washing protocols seem to play important role in the microbial contamination reduction. Therefore, the authors emphasize the risk of bacterial contamination of the household for both humans and pets, as well as the need for better communication of the current guidelines and the development of more feasible recommendations for long-term compliance. Overall, this manuscript is well written and well structured. The reviewer has a few questions to be addressed and also a few suggestions to consider for improvement:

1. The microbial counts are usually expressed as colony-forming units rather than cells in plating methods, because more than one cell may be present on the same spot to give rise to a single colony. Furthermore, there are bacteria that grow in chains and a single colony may represent several cells. Therefore, it is more appropriate to use total aerobic count.

2. Total aerobic count may refer to all the aerobic microorganisms present in the samples. But, not all microorganisms are able to form colonies in a culture medium. Further to the above comment, the reviewer suggests Aerobic plate count (APC) or even Total plate count (TPC).

3. It is mentioned in Lines 94-95 that ‘Owners were requested to complete one survey per dog with a maximum of two surveys per family’ and also in Lines 100-101 that ‘owners of 68 dogs (a total of 50 owners) were invited to complete a food bowl bacterial contamination study’. Why did the authors use survey data from the same dog owner, since it could be considered that the responses and their behaviour will be similar?

4. Line 134-135: Please describe the analysis in detail (which were the dependent and independent variables for the models). It is not so clear. In addition, Kruskall-Wallis can be used as an alternative to the Anova, when the assumption of normality or equality of variance is not met. Which assumption was not met? Where Kruskall-Wallis was used?

5. Figure 3: y-axis label: APC or TPC (log10 CFU/cm2), x-axis label: Treatment group. The mean values are presented in this graph? Please, report also what the error bars represent.

6. The reviewer believes that the report of the 95% confidence intervals for the statistically significant results would be a significant addition.

Other comments:

• Line 62: this study cultured for? Please, rephrase

• Line 120: 10 cm2 environmental sampling template (Name of Company, City, Country)

• Line 122: (Name of Company, City, Country) for Butterfields solution

• Line 124: Petrifilm TM aerobic count plates (Name of Company, City, Country)

• Line 219: Please, report the method for multiple testing correction?

• Line 130: colony forming units per cm squared (CFU/cm2)

• Figures 1 and 2: axis labels

• Please, report microbial counts with only one digit after the decimal point. Given the accuracy, one digit is sufficient.

6. PLOS authors have the option to publish the peer review history of their article (what does this mean?). If published, this will include your full peer review and any attached files.

Reviewer #1: No

Reviewer #2: No

---

## [Decision Letter · Decision Letter 1]

25 Feb 2022

PONE-D-21-32996R1Survey evaluation of dog owners’ feeding habits in a household setting and comparison of FDA hygiene protocols on dog bowl bacterial contamination as evaluated by total aerobic plate counts.PLOS ONE

Dear Dr. Luisana,

Thank you for submitting your manuscript to PLOS ONE. After careful consideration, we feel that it has merit but does not fully meet PLOS ONE’s publication criteria as it currently stands. Therefore, we invite you to submit a revised version of the manuscript that addresses the points raised during the review process.

ACADEMIC EDITOR:Beyond the comment raised by the reviewer regarding the revised manuscript, there are some points that according to the Academic Editor’s view should also be addressed by the authors prior to its consideration for publication in PLOS ONE. These additional comments raised by the Academic Editor are listed below: 1.
The title of the manuscript should be shorter and not so detailed; also, the period (.) at the end of the title should be deleted. Suggestion for title’s revision: “Survey evaluation of dog owners’ feeding practices and dog bowls’ hygiene assessment in domestic settings”. Corresponding revision for short title: “Evaluation of dog owners’ feeding and hygienic practices”. 2.
L32: NCSU-IACUC and -IRB should be spelled out 3.
L38, 39 and wherever else applicable: use past tense (e.g., had, stored, washed, prepared) when referring to results and protocols of the study. 4.
L41 (and throughout the manuscript): mentioning different p-values may get confusing for the reader; please refer consistently to the p-value decided to be treated as significance level in this study (P<0.05 for significant difference and P≥0.05 for non-significant differences) throughout the manuscript; there is no need for the specific attained p-value to be mentioned. 5.
 L45: there is no such thing as “high-risk” households but “high-risk populations” or “high-risk individuals” 6.
L49-50: this sentence should be revised for clarity 7.
L55: the phrase “the actual act of feeding” sounds weird…please revise to the simpler phrase “Pet feeding involves an interplay among the pet, the owner and the feed”. 8.
L65 and 66: correct to “Staphylococcus spp.” (“spp.” should not be italicized), “pseudomonads” (this is not a genus name but a bacterial group name and thus, it should not be italicized) and “Enterobacteriaceae” (family names should not be italicized). 9.
L75: revise to the “U.S. Food and Drug Administration” 10.
L81: use “implementation” in the place of “institution” 11.
L87, 101, 147 etc.: please delete the dash (-) after each subsection’s title 12.
L110-115: the sentence “Treatment group…wash and dry cycle” is rather long and inevitably complicated; please consider revising by splitting to at least two distinct sentences” 13.
L137: the phrase “due to percent change from 0 not being defined” could be omitted. 14.
L138-140: which linear models did you try? 15.
L156: please add a comma (,) prior to “whereas” 16.
L157: please mention the “less than 2% each” in parentheses comprehensively for the corresponding illnesses. 17.
L165: the “>25%” is not useful information, particularly when someone reads a phrase like “Lower levels of compliance”…either refer to a “<…” statement or provide a range for the corresponding values to give the reader a better idea of the magnitude 18.
Table 1: The "yes" or "no" statement is confusing when referring to compliance, where someone only "yes” would expect to be pertinent.  19.
Table 2: The presentation format could be improved with presenting first the answer, followed by the percentage in parentheses or after “:” 20.
L183-185: There is no need to repeat information in the text of the manuscript already provided in a tabular format (Table 2 in this particular case). Similarly in L190-192 and in L195. 21.
L199: please add a comma (,) prior to “respectively” 22.
L201: please revise the phrase “<1% and 3%, APC and overall, respectively” for clarity 23.
L206 and wherever else applicable: use “h” instead of hours” and “min” instead of “minutes” 24.
L209: change “between” to “among” 25.
L210: please add a period after “assessment” 26.
L217: please correct to “data were…”; “data” is the plural form of “datum” 27.
L219: please delete the period (.) after “APC” 28.
L225: the “non-significant difference” should be denoted by “p≥0.05” (and not by p<0.05); please correct accordingly 29.
L235: which linear regression model did you use 30.
L241: use directly the already spelled out “APC” 31.
L246: change “Whereas” to “On the other hand” 32.
L251: which statistical model do you refer to? 33.
L259: correct to “…and did not follow” 34.
L272: revise to “According to Weese et al. (2010), dog bowls were 17 times more likely…” 35.
L282: please use “efficacy of sanitation” instead of “degree of sanitation” 36.
L292: raw meat cannot be infected but contaminated 37.
L294-296: the sentence “The effects of cross-contamination may extend beyond bacterial contamination when one considers that 9% of pet owners reported adding medications or supplements into their pets’ food bowls.” is not clear; please revise. 38.
L299: “…may have implications as far as increased microbial risk”: it is not clear what you mean to say here 39.
L307: veterinarians do not have consumers…most likely, “clients” is a more appropriate word here 40.
L309: use “allow for” instead of “institute” 41.
L315: microbial growth was not assessed in the context of this study 42.
L320: use “establishment” or “development” or “implementation” instead of “institution”.

We look forward to receiving your revised manuscript.

Kind regards,

Alexandra Lianou, M.Sc., Ph.D.

Academic Editor

PLOS ONE

Journal Requirements:

Reviewers' comments:

Reviewer's Responses to Questions

**Comments to the Author**

1. If the authors have adequately addressed your comments raised in a previous round of review and you feel that this manuscript is now acceptable for publication, you may indicate that here to bypass the “Comments to the Author” section, enter your conflict of interest statement in the “Confidential to Editor” section, and submit your "Accept" recommendation.

Reviewer #2: All comments have been addressed

2. Is the manuscript technically sound, and do the data support the conclusions?

Reviewer #2: Yes

3. Has the statistical analysis been performed appropriately and rigorously? 

Reviewer #2: Yes

4. Have the authors made all data underlying the findings in their manuscript fully available?

Reviewer #2: Yes

5. Is the manuscript presented in an intelligible fashion and written in standard English?

Reviewer #2: Yes

6. Review Comments to the Author

Reviewer #2: Please, check again the y-axis label for Figure 3. The correct is APC (log CFU/cm^2). Also add in the figure legend that the results were expressed as mean (± standard error of the mean, SEM) log CFU/cm^2.

7. PLOS authors have the option to publish the peer review history of their article (what does this mean?). If published, this will include your full peer review and any attached files.

Reviewer #2: No

---

## [Author Response · Author response to Decision Letter 1]

2 Mar 2022

Please see attached letter for reply to revision requests.

---

## [Editor Report · Decision Letter 2]

9 Mar 2022

Survey evaluation of dog owners’ feeding practices and dog bowls’ hygiene assessment in domestic settings

PONE-D-21-32996R2

Dear Dr. Luisana,

We’re pleased to inform you that your manuscript has been judged scientifically suitable for publication and will be formally accepted for publication once it meets all outstanding technical requirements.

Kind regards,

Alexandra Lianou, M.Sc., Ph.D.

Academic Editor

PLOS ONE
---

## [Editor Report · Acceptance letter]

14 Mar 2022

PONE-D-21-32996R2 

Survey evaluation of dog owners’ feeding practices and dog bowls’ hygiene assessment in domestic settings 

Dear Dr. Luisana:

I'm pleased to inform you that your manuscript has been deemed suitable for publication in PLOS ONE. Congratulations! Your manuscript is now with our production department. 

Kind regards, 

on behalf of

Dr. Alexandra Lianou 

Academic Editor

PLOS ONE